
## Oceanic and atmospheric methane cycling in the cGENIE Earth system model

Christopher T. Reinhard[1,2,3*], Stephanie L. Olson[2,4,5], Sandra Kirtland Turner[6], Cecily Pälike[7], Yoshiki Kanzaki[6], Andy Ridgwell[6]

[1]School of Earth and Atmospheric Sciences, Georgia Institute of Technology, Atlanta, GA 30332
[2]NASA Astrobiology Institute, Alternative Earths Team, Riverside, CA
[3]NASA Nexus for Exoplanet System Science (NExSS) Upside-Down Biospheres Team, Georgia Institute of Technology, Atlanta, GA
[4]Department of Geophysical Sciences, University of Chicago, Chicago, IL 60637
[5]Department of Earth, Atmospheric, and Planetary Science, Purdue University, West Lafayette, IN 47907
[6]Department of Earth Sciences, University of California, Riverside, Riverside, CA 92521
[7]MARUM Center for Marine Environmental Sciences, University of Bremen, Germany

*To whom correspondence should be addressed. E-mail: chris.reinhard@eas.gatech.edu

**Abstract: The methane ($CH_4$) cycle is a key component of the Earth system that links planetary climate, biological metabolism, and the global biogeochemical cycles of carbon, oxygen, sulfur, and hydrogen. However, currently lacking is a numerical model capable of simulating a diversity of environments in the ocean where $CH_4$ can be produced and destroyed, and with the flexibility to be able to explore not only relatively recent perturbations to Earth's $CH_4$ cycle but also to probe $CH_4$ cycling and associated climate impacts under the very low-$O_2$ conditions characteristic of most of Earth history and likely widespread on other Earth-like planets. Here, we present a refinement and expansion of the ocean-atmosphere $CH_4$ cycle in the intermediate-complexity Earth system model cGENIE, including parameterized atmospheric $O_2$-$O_3$-$CH_4$ photochemistry and schemes for microbial methanogenesis, aerobic methanotrophy, and anaerobic oxidation of methane (AOM). We describe the model framework, compare model parameterizations against modern observations, and illustrate the flexibility of the model through a series of example simulations. Though we make no attempt to rigorously tune default model parameters, we find that simulated atmospheric $CH_4$ levels and marine dissolved $CH_4$ distributions are generally in good agreement with empirical constraints for the modern and recent Earth. Finally, we illustrate the model's utility in understanding the time-dependent behavior of the $CH_4$ cycle resulting from transient carbon injection into the atmosphere, and present model ensembles that examine the effects of atmospheric $pO_2$, oceanic dissolved $SO_4^{2-}$, and the thermodynamics of microbial metabolism on steady-state atmospheric $CH_4$ abundance. Future model developments will address the sources and sinks of $CH_4$ associated with the terrestrial biosphere and marine $CH_4$ gas hydrates, both of which will be essential for comprehensive treatment of Earth's $CH_4$ cycle during geologically recent time periods.**

## 1. Introduction

The global biogeochemical cycle of methane ($CH_4$) is central to the evolution and climatic stability of the Earth system. Methane provides an important substrate for microbial metabolism, particularly in energy-limited microbial ecosystems in the deep subsurface (Valentine, 2011;Chapelle et al., 1995) and in anoxic marine and lacustrine sediments (Lovley et al.,



1982;Hoehler et al., 2001). Indeed, the microbial production and consumption of $CH_4$ are amongst
the oldest metabolisms on Earth, with an isotopic record of bacterial methane cycling stretching
back nearly 3.5 billion years (Ueno et al., 2006;Hinrichs, 2002;Hayes, 1994). As the most abundant
hydrocarbon in Earth's atmosphere $CH_4$ also has a significant influence on atmospheric
photochemistry (Thompson and Cicerone, 1986), and because it absorbs in a window region of
Earth's longwave emission spectrum it is an important greenhouse gas. This has important
implications over the coming centuries, with atmospheric $CH_4$ classified as a critical near-term
climate forcing (Myhre et al., 2013), but has also resulted in dramatic impacts during certain
periods of Earth history. For example, high steady-state atmospheric $CH_4$ has been invoked as an
important component of Earth's early energy budget, potentially helping to offset a dim early Sun
(Sagan and Mullen, 1972;Pavlov et al., 2000;Haqq-Misra et al., 2008), while time-dependent
changes to the atmospheric $CH_4$ inventory have been invoked as drivers of extreme climatic
perturbations throughout Earth history (Dickens et al., 1997;Dickens, 2003;Bjerrum and Canfield,
2011;Zeebe, 2013;Schrag et al., 2002). Because it is cycled largely through biological processes
on the modern (and ancient) Earth and is spectrally active, atmospheric $CH_4$ has also been
suggested as a remotely detectable biosignature that could be applied to planets beyond our solar
system (Hitchcock and Lovelock, 1967;Sagan et al., 1993;Krissansen-Totton et al., 2018).

A number of low-order Earth system models incorporating a basic $CH_4$ cycle have been developed,
particularly with a view to addressing relatively 'deep time' geological questions. These include
explorations of long-term changes to the chemistry of Earth's atmosphere (Claire et al.,
2006;Catling et al., 2007;Bartdorff et al., 2008;Beerling et al., 2009), potential climate impacts at
steady state (Kasting et al., 2001;Ozaki et al., 2018), and transient impacts of $CH_4$ degassing on
climate (Schrag et al., 2002;Bjerrum and Canfield, 2011). In some cases these models explicitly
couple surface fluxes to a model of atmospheric photochemistry (Lamarque et al., 2006;Ozaki et
al., 2018;Kasting et al., 2001), but in general atmospheric chemistry is parameterized based on
offline 1- or 2-D photochemical models while surface fluxes are specified arbitrarily or are based
on a simple 1-box ocean-biosphere model. A range of slightly more complex 'box' model
approaches have been applied to simulate transient perturbations to Earth's $CH_4$ cycle and
attendant climate impacts on timescales ranging from $\sim 10^5$ years (Dickens et al., 1997;Dickens,
2003) to $\sim 10^8$ years (Daines and Lenton, 2016). In addition, offline and/or highly parameterized





approaches toward simulating the impact of transient $CH_4$ degassing from gas hydrate reservoirs
have been developed and applied to relatively recent periods of Earth history (Archer and Buffett,
2005;Lunt et al., 2011) or projected future changes (Archer et al., 2009;Hunter et al., 2013).
However, the most sophisticated and mechanistic models of global $CH_4$ cycling currently available
tend to focus on terrestrial (soil or wetland) sources and sinks (Ridgwell et al., 1999;Walter and
Heimann, 2000;Wania et al., 2010;Konijnendijk et al., 2011;Melton et al., 2013) or focus on
explicitly modeling atmospheric photochemistry (Shindell et al., 2013).

Much less work has been done to develop ocean biogeochemistry models that are both equipped
to deal with the wide range of boundary conditions characteristic of Earth history and are
computationally tractable when running large model ensembles and/or on long (approaching $\sim 10^6$
year) timescales, as well as being able to simulate the (3-D) redox structure of the ocean allowing
for localized zones of production and oxidation (which provides more accurate estimates of
emission to the atmosphere). For instance, Elliot et al. (2011) advanced modelling of marine $CH_4$
cycling by developing and employing a 3-D ocean circulation and climate model (CCSM-3) to
simulate the impact of injecting clathrate-derived $CH_4$ into the Arctic ocean. However, microbial
consumption of $CH_4$ in the ocean interior was parameterized via an empirical log-linear function
that implicitly neglects anaerobic oxidation of methane (AOM) via dissolved sulfate ($SO_4^{2-}$), which
on the modern Earth is an enormously important internal $CH_4$ sink within Earth's oceans (Egger
et al., 2018). Their simulations did not explore atmospheric chemistry. Similarly, Daines and
Lenton (2016) also innovated over traditional box modelling approaches by applying an ocean
general circulation model (GCM) to examine the role of aerobic methanotrophy in modulating
ocean-atmosphere fluxes of $CH_4$ during Archean time (prior to $\sim$2.5 billion years ago, Ga).
However, this analysis likewise did not include AOM, and the GCM results were not coupled to
atmospheric chemistry. In contrast, Olson et al. (2016) included AOM in a 3-D ocean
biogeochemistry model coupled to an atmospheric chemistry routine and found that AOM
represents a critical internal $CH_4$ sink in the oceans even at relatively low dissolved $SO_4^{2-}$ levels.
Though this represented an important further step forward in understanding marine $CH_4$ cycling
on the early Earth, Olson et al. (2016) employed a simplified parameterization of aerobic $CH_4$
consumption, neglected the thermodynamics of $CH_4$-consuming metabolisms under energy-
limited conditions, and employed a parameterization of atmospheric $O_2$-$O_3$-$CH_4$ photochemistry



that is most readily applicable to only a subset of the atmospheric $pO_2$ values characteristic of
Earth history (Daines and Lenton, 2016;Olson et al., 2016). While all of these studies provided
new modelling innovations and advances in understanding, important facets of global $CH_4$ cycling,
particularly as relevant to the evolution of early Earth, were lacking.

Here, we present a new framework for modeling the ocean-atmosphere biogeochemical $CH_4$ cycle
in the 'muffin' release of the cGENIE Earth system model. Our goal is to make further progress
in the development of a flexible intermediate-complexity model suitable for simulating the global
biogeochemical $CH_4$ cycle on ocean-bearing planets, with an initial focus on periods of Earth
history (or other habitable ocean-bearing planets) that lack a robust terrestrial biosphere. We also
aim to provide a numerical modeling foundation from which to further develop a more complete
$CH_4$ cycle within the cGENIE framework, including, for example, dynamic $CH_4$ hydrate cycling
and the production/consumption of $CH_4$ by terrestrial ecosystems.

The outline of the paper is as follows. In Section 2 we briefly describe the GENIE/cGENIE Earth
system model, with a particular eye toward the features that are most relevant for the biological
carbon pump and the oceanic $CH_4$ cycle. In Section 3 we describe the major microbial metabolisms
involved in the oceanic $CH_4$ cycle and compare our parameterizations to data from modern marine
environments. In Section 4 we describe two alternative parameterizations of atmospheric $O_2$-$O_3$-
$CH_4$ photochemistry incorporated into the model and compare these to modern/recent
observations. In Section 5 we present results from a series of idealized simulations meant to
illustrate the flexibility of the model and some potential applications. The availability of the model
code, plus configuration files for all experiments described in the paper, is provided in Section 7,
following a brief summary in Section 6.

**2. The GENIE/cGENIE Earth system model**
**2.1. Ocean physics and climate model – C-GOLDSTEIN**
The ocean physics and climate model in cGENIE is comprised of a reduced physics (frictional
geostrophic) 3-D ocean circulation model coupled to both a 2-D energy-moisture balance model
(EMBM) and a dynamic-thermodynamic sea-ice model (Edwards and Marsh, 2005;Marsh et al.,
2011). The ocean model transports heat, salinity, and biogeochemical tracers using a scheme of





parameterized isoneutral diffusion and eddy-induced advection (Griffies, 1998;Edwards and
Marsh, 2005;Marsh et al., 2011), exchanges heat and moisture with the atmosphere, sea ice, and
land, and is forced at the ocean surface by the input of zonal and meridional wind stress according
to a specified wind field. The 2-D atmospheric energy-moisture-balance model (EMBM) considers
the heat and moisture balance for the atmospheric boundary layer using air temperature and
specific humidity as prognostic tracers. Heat and moisture are mixed horizontally throughout the
atmosphere, and exchange heat and moisture with the ocean and land surfaces with precipitation
occurring above a given relative humidity threshold. The sea-ice model tracks the horizontal
transport of sea ice, and the exchange of heat and freshwater with the ocean and atmosphere using
ice thickness, areal fraction, and concentration as prognostic variables. Full descriptions of the
model and coupling procedure can be found in Edwards and Marsh (2005) and, more recently, in
Marsh et al. (2011). As implemented here, the ocean model is configured as a 36 x 36 equal-area
grid (uniform in longitude and uniform in the sine of latitude) with 16 logarithmically spaced depth
levels and seasonal surface forcing from the EMBM.

**2.2. Ocean biological pump – BIOGEM**
The biogeochemical model component — 'BIOGEM' — regulates air-sea gas exchange as well
as the transformation and partitioning of biogeochemical tracers within the ocean, as described in
Ridgwell et al. (2007). By default, the biological pump is driven by parameterized uptake of
nutrients in the surface ocean, with this flux converted stoichiometrically to biomass that is then
partitioned into either dissolved or particulate organic matter for downstream transport, sinking,
and remineralization. Dissolved organic matter is transported by the ocean model and decays with
a specified time constant, while particulate organic matter is immediately exported out of the
surface ocean and partitioned into two fractions of differing lability. In the ocean interior,
particulate organic matter is remineralized instantaneously throughout the water column following
an exponential decay function with a specified remineralization length scale.

In the simulations discussed below, photosynthetic nutrient uptake in surface ocean grid cells is
controlled by a single limiting nutrient, dissolved phosphate ($PO_4$):

$$\frac{\partial PO_4}{\partial t} = -\Gamma + \lambda DOP, \tag{1}$$





$$\frac{\partial \mathrm{DOP}}{\partial t} = \upsilon\Gamma - \lambda\mathrm{DOP} , \tag{2}$$

where DOP represents dissolved organic phosphorus, $\upsilon$ represents the proportion of photosynthetic
production that is initially partitioned into a dissolved organic phase, $\lambda$ represents a decay constant
(time$^{-1}$) for dissolved organic matter, and $\Gamma$ represents photosynthetic nutrient uptake following
Doney et al. (2006):

$$\Gamma = F_I \cdot F_N \cdot F_T \cdot (1 - f_{ice}) \cdot \frac{[\mathrm{PO}_4]}{\tau_{bio}}. \tag{3}$$

Rates of photosynthesis are regulated by terms describing the impact of available light ($F_I$),
nutrient abundance ($F_N$), temperature ($F_T$), and fractional sea ice coverage in each grid cell ($f_{ice}$).
Rates of photosynthetic nutrient uptake are further scaled to ambient dissolved $PO_4$ ($[PO_4]$)
according to an optimal uptake timescale ($\tau_{bio}$).

Note that this parameterization differs from that in Ridgwell et al. (2007). Specifically, the impacts
of light and nutrient availability are both described via Michaelis-Menten terms:

$$F_I = \frac{I}{I + \kappa_I} , \tag{4}$$

$$F_N = \frac{[\mathrm{PO}_4^{3-}]}{\kappa_P + [\mathrm{PO}_4^{3-}]} , \tag{5}$$

where shortwave irradiance $I$ is averaged over the entire mixed layer, and is assumed to decay
exponentially from the sea surface with a length scale of 20 m. It is assumed that nutrient uptake
and photosynthetic production only occur in surface grid cells of cGENIE (e.g., the upper 80 m),
which is similar to the 'compensation depth' $z_c$ in Doney et al. (2006) of 75 m. The terms $\kappa_I$ and
$\kappa_P$ represent half-saturation constants for light and dissolved phosphate, respectively. In addition,
the effect of temperature on nutrient uptake is parameterized according to:

$$F_T = k_{T0} \cdot \exp\left[\frac{T}{k_{eT}}\right] , \tag{6}$$

where $k_{T0}$ and $k_{eT}$ denote pre-exponential and exponential scaling constants and $T$ represents
absolute *in-situ* temperature. The scaling constants are chosen to give approximately a factor of
two change in rate with a temperature change of 10ºC (e.g., a $Q_{10}$ response of ~2.0). Lastly, the



final term in Eq. (3), not present in the default parameterization of Ridgwell et al. (2007), allows
for biological productivity to scale more directly with available $PO_4$ when dissolved $PO_4$
concentrations are elevated relative to those of the modern oceans.

Particulate organic matter (POM) is immediately exported out of the surface ocean without lateral
advection, and is instantaneously remineralized throughout the water column according to an
exponential function of depth:

$$F_z^{POM} = F_{z=z_h}^{POM} \cdot \left( \sum_i r_i^{POM} \cdot \exp\left( \frac{z_h - z}{l_i^{POM}} \right) \right) , \tag{7}$$

where $F_z^{POM}$ is the particulate organic matter flux at a given depth (and $z_h$ is the base of the photic
zone), $z$ is depth, $r_i^{POM}$ and $l_i^{POM}$ refer to the relative partitioning into each organic matter lability
fraction $i$ and the $e$-folding depth of that fraction, respectively. The simulations presented here
employ two organic matter fractions, a 'labile' fraction (94.5%) with an $e$-folding depth of ~590
m and an effectively inaccessible fraction (5.5%) with an $e$-folding depth of $10^6$ m (**Table 1**).

We employ a revised scheme for organic matter remineralization in the ocean interior, following
that commonly used in models of organic matter remineralization within marine and lacustrine
sediments (Rabouille and Gaillard, 1991;Van Cappellen et al., 1993;Boudreau, 1996a, b).
Respiratory electron acceptors ($O_2$, $NO_3^-$, and $SO_4^{2-}$) are consumed according to decreasing free
energy yield (Froelich et al., 1979), with consumption rates ($R_i$) scaled to both electron acceptor
abundance and the inhibitory impact of electron acceptors with higher intrinsic free energy yield:

$$R_{O_2} = \frac{[O_2]}{\kappa_{O_2} + [O_2]} , \tag{8}$$

$$R_{NO_3} = \frac{[NO_3]}{\kappa_{NO_3} + [NO_3]} \cdot \frac{\kappa_{O_2}^i}{\kappa_{O_2}^i + [O_2]} , \tag{9}$$

$$R_{SO_4} = \frac{[SO_4]}{\kappa_{SO_4} + [SO_4]} \cdot \frac{\kappa_{O_2}^i}{\kappa_{O_2}^i + [O_2]} \cdot \frac{\kappa_{NO_3}^i}{\kappa_{NO_3}^i + [NO_3]} , \tag{10}$$

with the exception that in the biogeochemical configuration used here we do not consider nitrate
($NO_3^-$). The total consumption of settling POM within each ocean layer is governed by the
predetermined remineralization profiles (Equation 7). The $R_i$ terms denote the relative fraction of



this organic matter consumption that is performed by each respiratory process. We specify a closed
system with no net organic matter burial in marine sediments (see below) and hence the POM flux
to the sediment surface is assumed to be completely degraded, with the same partitioning amongst
electron acceptors carried out according to local bottom water chemistry. For DOM, the assumed
lifetime ($\lambda$) determines the total fraction of DOM degraded (and Equations 8-10 again determine
how the consumption of electron acceptors is partitioned). The $\kappa_i$ terms represent half-saturation
constants for each metabolism, $\kappa_i^i$ terms give inhibition constants acting on less energetic
downstream respiratory processes, and brackets denote concentration. Default parameter values
used here are shown in **Table 1**.

**3. Oceanic methane cycling**
**3.1. Microbial methanogenesis**
Methanogenesis represents the terminal step in our remineralization scheme, and follows the
overall stoichiometry:

$$2CH_2O \rightarrow CH_4 + CO_2 \ .$$

This can be taken to implicitly include fermentation of organic matter to acetate followed by
acetoclastic methanogenesis:

$$2CH_2O \rightarrow CH_3COOH \ ,$$

$$CH_3COOH \rightarrow CH_4 + CO_2 \ ,$$

or the fermentation of organic matter to acetate followed by anaerobic acetate oxidation and
hydrogenotrophic methanogenesis:

$$2CH_2O \rightarrow CH_3COOH \ ,$$

$$CH_3COOH + 2H_2O \rightarrow 4H_2 + 2CO_2 \ ,$$

$$4H_2 + CO_2 \rightarrow CH_4 + 2H_2O \ ,$$

both of which have the same overall net stoichiometry provided that $H_2$ is assumed to be
quantitatively converted to $CH_4$ by hydrogenotrophic methanogens. We thus ignore the scenario
in which some fraction of $H_2$ is converted directly to biomass by hydrogenotrophic methanogens
acting as primary producers (Ozaki et al., 2018).




Because we specify a closed system with no net organic matter burial in marine sediments, all
organic matter not remineralized by more energetic respiratory metabolisms is converted into $CH_4$
(e.g., $R_{CH4} = 1 - R_{O2} - R_{NO3} - R_{SO4}$):

$$R_{CH_4} = \frac{\kappa^i_{O_2}}{\kappa^i_{O_2} + [O_2]} \cdot \frac{\kappa^i_{NO_3}}{\kappa^i_{NO_3} + [NO_3]} \cdot \frac{\kappa^i_{SO_4}}{\kappa^i_{SO_4} + [SO_4]}, \tag{11}$$

where $\kappa_i$ and $\kappa^i_i$ terms are as described above (**Table 1**). We disable nitrate ($NO_3$) as a tracer in the
simulations presented here, such that anaerobic remineralization of organic matter is partitioned
entirely between sulfate reduction and methanogenesis (**Fig. 1**). Using our default parameter
values (**Table 1**), aerobic respiration dominates organic matter remineralization at $[O_2]$ values
significantly above 1 μmol kg$^{-1}$ (**Fig. 1a**) while anaerobic remineralization is dominated by
methanogenesis at $[SO_4^{2-}]$ values significantly below 1 mmol kg$^{-1}$ (**Fig. 1b**). An important outcome
of the revised 'inhibition' scheme is that metabolic pathways with differing intrinsic free energy
yields can coexist, which more accurately reflects field observations from a range of natural
settings (Curtis, 2003;Bethke et al., 2008;Kuivila et al., 1989;Jakobsen and Postma, 1999). In
particular, it allows us to roughly capture the impact of oxidant gradients within sinking marine
aggregates (Bianchi et al., 2018), which can facilitate non-trivial anaerobic carbon
remineralization within sinking particles even in the presence of relatively high $[O_2]$ in the ocean
water column (**Fig. 1c**).

While the model tracks the carbon isotope composition of oceanic and atmospheric $CH_4$ ($\delta^{13}C$,
reported in per mil notation relative to the Pee Dee Belemnite, PDB), the only significant isotope
effect we include here is that attendant to acetoclastic methanogenesis. We specify a constant
isotope fractionation between organic carbon and $CH_4$ during methanogenesis of -35‰ by default
(**Table 2**), which will tend to produce microbial $CH_4$ with a $\delta^{13}C$ composition of roughly -60‰
when combined with the default isotope fractionation associated with photosynthetic carbon
fixation in the surface ocean (e.g., Kirtland Turner and Ridgwell, 2016). The model does not
currently include any potential isotope effects associated with aerobic/anaerobic methanotrophy,
air-sea gas exchange of $CH_4$, or photochemical breakdown of $CH_4$ in the atmosphere. It does,
however, include a comprehensive $^{13}C$ scheme associated with ocean-atmosphere cycling of $CO_2$
(Kirtland Turner and Ridgwell, 2016;Ridgwell, 2001).





**3.2. Aerobic methanotrophy**
Microbial aerobic methanotrophy proceeds according to:

$$CH_4 + 2O_2 \rightarrow CO_2 + 2H_2O$$

This reaction is highly favorable energetically, with a free energy yield under standard conditions
of ~850 kJ per mole of methane consumed (**Table 2**). We represent rates of aerobic methanotrophy
($R_{AER}$) with a mixed kinetic-thermodynamic formulation (Jin & Bethke, 2005; 2007; Regnier et
al., 2011), in which $CH_4$ oxidation kinetics are controlled by substrate availability, thermodynamic
energy yield, and temperature:

$$R_{AER} = k_{AER} \cdot F_k^{AER} \cdot F_t^{AER} \cdot F_T .$$ (12)

A rate constant for aerobic methanotrophy ($y^{-1}$) is defined as $k_{AER}$, while $F_i$ terms denote kinetic
($k$) and thermodynamic ($t$) factors as defined below and a temperature ($T$) factor as given in Eq.
(6) above.

The kinetic factor ($F_k$) for aerobic methanotrophy is controlled by substrate availability according
to:

$$F_k^{AER} = [CH_4] \cdot \frac{[O_2]}{\kappa_O^{AER} + [O_2]} ,$$ (13)

where brackets denote concentration and the $\kappa$ term denotes a half-saturation constant with respect
to $O_2$. We employ a hybrid parameterization in which kinetics are first-order with respect to $CH_4$
but also scaled by a Michaelis-Menten-type term for $O_2$. This formulation is based on the rationale
that half-saturation constants for $CH_4$ are typically similar to (or greater than) the dissolved $CH_4$
levels attained in anoxic water column environments (Regnier et al., 2011) but is also meant to
allow for rapid $CH_4$ consumption under 'bloom' conditions with an appropriately scaled rate
constant (see below).

The effect of thermodynamic energy yield on aerobic methanotrophy is given by:

$$F_t^{AER} = 1 - \exp\left[ \frac{\Delta G_{r,AER} + \Delta G_{BQ,AER}}{\chi RT} \right],$$ (14)

where $\Delta G_r$ denotes the Gibbs free energy of reaction under *in-situ* conditions, $\Delta G_{BQ}$ represents the
minimum energy required to sustain ATP synthesis (Hoehler et al., 2001;Hoehler, 2004;Jin and



Bethke, 2007), $\chi$ is the stoichiometric number of the reaction (e.g., the number of times the rate-
determining step occurs in the overall process), and $R$ and $T$ represent the gas constant and absolute
*in-situ* temperature, respectively. The available free energy is estimated according to:

$$\Delta G_{r,AER} = \Delta G_{r,AER}^0 + RT \cdot \ln \frac{\gamma_{CO_2}[CO_2]}{\gamma_{O_2}[O_2] \cdot \gamma_{CH_4}[CH_4]} \, , \tag{15}$$

where, in addition to the terms defined above, $\Delta G_r^0$ represents the Gibbs free energy of the reaction
under standard conditions, and $\gamma_i$ values represent activity coefficients. Note that we assume an
$H_2O$ activity of unity.

**3.3. Anaerobic oxidation of methane (AOM)**
The oxidation of methane can also be coupled to electron acceptors other than $O_2$, including nitrate
($NO_3^-$), sulfate ($SO_4^{2-}$), and oxide phases of iron (Fe) and manganese (Mn) (Reeburgh,
1976;Martens and Berner, 1977;Hoehler et al., 1994;Hinrichs et al., 1999;Orphan et al.,
2001;Sivan et al., 2011;Haroon et al., 2013;Egger et al., 2015). Because it is by far the most
abundant of these oxidants on the modern Earth, and has likely been the most abundant throughout
Earth's history, we focus on anaerobic oxidation of methane (AOM) at the expense of $SO_4^{2-}$:

$$CH_4 + SO_4^{2-} \rightarrow HCO_3^- + HS^- + H_2O \, .$$

This process is currently thought to be performed most often through a syntrophic association
between Archaea and sulfate reducing bacteria (Boetius et al., 2000), though the mechanics
controlling the exchange of reducing equivalents within the syntrophy remain to be fully elucidated
(Milucka et al., 2012;McGlynn et al., 2015). In any case, consumption of $CH_4$ at the sulfate-
methane transition zone (SMTZ) represents an extremely large sink flux of $CH_4$ in modern marine
sediments (Regnier et al., 2011;Egger et al., 2018).

Anaerobic methanotrophy is much less energetically favorable under standard conditions, with a
free energy yield of ~30 kJ per mole of $CH_4$ (**Table 2**). As a result, the influence of
thermodynamics on rates of AOM is potentially much stronger than it will tend to be in the case
of aerobic methanotrophy. As above, rates of AOM are controlled by the combined influence of
substrate availability, thermodynamic drive, and temperature:

$$R_{AOM} = k_{AOM} \cdot F_k^{AOM} \cdot F_t^{AOM} \cdot F_T \tag{16}$$





where $k_{AOM}$ is a rate constant for anaerobic methane oxidation (y⁻¹), while $F_i$ terms denote kinetic
($k$) and thermodynamic ($t$) factors as defined below and a temperature ($T$) factor as given in Eq.
(6) above.

The kinetics of anaerobic methane oxidation are specified according to:

$$F_k^{AOM} = [CH_4] \cdot \frac{[SO_4^{2-}]}{\kappa_S^{AOM} + [SO_4^{2-}]}$$    (17)

where brackets denote concentration and the $\kappa$ term denotes a half-saturation constant with respect
to $SO_4^{2-}$. We employ a hybrid parameterization in which kinetics are first-order with respect to
$CH_4$ but are also scaled by a Michaelis-Menten-type term for $SO_4^{2-}$ for reasons discussed above.

The effect of thermodynamic energy yield on anaerobic methane oxidation is specified as follows:

$$F_t^{AOM} = 1 - \exp\left[ \frac{\Delta G_{r,AOM} + \Delta G_{BQ,AOM}}{\chi RT} \right] \quad .$$    (18)

As above, $\Delta G_r$ denotes the Gibbs free energy of reaction under *in-situ* conditions, $\Delta G_{BQ}$ is the
minimum energy required to sustain ATP synthesis (the 'biological quantum'), $\chi$ is the
stoichiometric number of the reaction, and $R$ and $T$ represent the gas constant and absolute *in-situ*
temperature, respectively. The available free energy for AOM under *in-situ* conditions is estimated
according to:

$$\Delta G_{r,AOM} = \Delta G_{r,AOM}^0 + RT \cdot \ln \frac{\gamma_{HCO_3^-}[HCO_3^-] \cdot \gamma_{HS^-}[HS^-]}{\gamma_{SO_4^{2-}}[SO_4^{2-}] \cdot \gamma_{CH_4}[CH_4]} \quad ,$$    (19)

where $\Delta G_r^0$ again represents the Gibbs free energy of the net AOM reaction given above under
standard conditions, and $\gamma_i$ values represent activity coefficients. Again, we assume an $H_2O$
activity of unity.

**3.4. Default parameters for aerobic and anaerobic methanotrophy**
We choose default rate constants according to a dataset of compiled rates of aerobic and anaerobic
methanotrophy in oxygenated and anoxic marine water column environments (see Supplementary
Data), after correction to *in-situ* temperature (**Fig. 2a, b**). Our default values for both rate constants
are on the low end of the observational dataset, but are very roughly tuned to yield steady-state



diffusive CH$_4$ fluxes from the ocean that are consistent with recent observational constraints (**Fig.**
**2c**). It is important to note, however, that these values are not extensively tuned and could be
adjusted depending on the application. For example, transient CH$_4$ release experiments could
employ rate constants that are scaled upward to reflect transient ('bloom') elevations in microbial
community CH$_4$ consumption as observed in field studies (Kessler et al., 2011;Crespo-Medina et
al., 2014). Default values for other kinetic parameters (**Table 2**) are chosen to be broadly consistent
with field measurements and pure/mixed culture experiments with aerobic methanotrophs (Bender
and Conrad, 1992, 1993;Hanson and Hanson, 1996;Dunfield and Conrad, 2000;van Bodegom et
al., 2001), and to remain roughly consistent with previous work for comparative purposes (e.g.,
Olson et al., 2016), though the parameters have not been formally tuned and we explore model
sensitivity below.

Thermodynamic energy yields of each reaction under standard conditions are calculated based on
the standard molal thermodynamic properties given in Regnier et al. (2011). Stoichiometric
numbers are assumed identical for both metabolisms, with default values of 1.0 (Jin and Bethke,
2005;Dale et al., 2006). We assume a default biological quantum ($\Delta G_{BQ}$) of 15 kJ mol$^{-1}$ for both
aerobic and anaerobic methanotrophy, though these can be expected to vary somewhat as a
function of metabolism and environmental conditions (Schink, 1997;Hoehler, 2004;Dale et al.,
2008). These can be varied independently for aerobic and anaerobic methanotrophy in the model,
and we explore model sensitivity to this parameter below. Lastly, for simplicity and to minimize
computational expense we assume constant activity coefficients for each species throughout all
ocean grid cells (**Table 2**). For some applications it may ultimately be important to add a scheme
for estimating activity coefficients according to ambient salinity and ion chemistry, for example
estimating methane fluxes in planetary scenarios with very different major ion chemistry or much
higher/lower salinity than those characteristic of Earth's modern oceans.

**4. Atmospheric methane cycling**
**4.1. Air-sea gas exchange**
Ocean-atmosphere fluxes of CH$_4$ ($J_{gas}$) are governed by temperature- and salinity-dependent
solubility and surface wind speed above a given grid cell:





$$J_{gas} = A \cdot k_{gas} \cdot \left([CH_4]_{sat} - [CH_4]_{cell}\right) \quad , \tag{20}$$

where $A$ denotes the area available for gas-exchange (e.g., the area of ice-free surface ocean),
$[CH_4]_{cell}$ denotes the ambient dissolved $CH_4$ concentration in a given surface ocean grid cell,
$[CH_4]_{sat}$ represents the dissolved $CH_4$ concentration at saturation with a given atmospheric $pCH_4$,
temperature, and salinity, and $k_{gas}$ represents a gas transfer velocity. Solubility is based on a Bunsen
solubility coefficient ($\beta$) corrected for ambient temperature ($T$) and salinity ($S$) according to:

$$\ln \beta = a_1 + a_2(100/T) + a_3 \ln(T/300) + S\left[b_1 + b_2(T/100) + b_3(T/100)^2\right], \tag{21}$$

[Note that the Henry's law constant $K_0$ is related to the Bunsen solubility coefficient by $K_0 =$
$\beta/\rho V^+$, where $\rho$ is density and $V^+$ is the molar volume of the gas at STP.] Gas transfer velocity
($k_{gas}$) is calculated based on the surface windspeed ($u$) and a Schmidt number (Sc) corrected for
temperature assuming a constant salinity of 35‰:

$$k_{gas} = k \cdot u^2 \cdot \left[Sc/660\right]^{-0.5}, \tag{22}$$

where $k$ is a dimensionless gas transfer coefficient, $u$ is surface wind speed, and Sc is the
temperature-corrected Schmidt number according to:

$$Sc = c_1 - c_2 T + c_3 T^2 - c_4 T^3. \tag{23}$$

All default constants and coefficients for the gas exchange scheme are given in **Table 3**. Overall,
the scheme for air-sea gas exchange of $CH_4$ follows by default that for other gases accounted for
in BIOGEM, such as $O_2$ and $CO_2$, as described in (Ridgwell et al., 2007)

**4.2. Parameterized $O_2$-$O_3$-$CH_4$ photochemistry**
Once degassed to the atmosphere, $CH_4$ becomes involved in a complex series of photochemical
reactions initiated by hydroxyl radical (OH) attack on $CH_4$ (Kasting et al., 1983;Prather,
1996;Pavlov et al., 2000;Schmidt and Shindell, 2003). Following Claire et al. (2006) and Goldblatt
et al. (2006), we parameterize $O_2$-$O_3$-$CH_4$ photochemistry according to a bimolecular 'rate law':

$$J_{CH_4} = k_{eff} \cdot M_{O_2} \cdot M_{CH_4}, \tag{24}$$

where $M_i$ terms represent the atmospheric inventories of $O_2$ and $CH_4$, respectively, and $k_{eff}$ denotes
an effective rate constant ($Tmol^{-1} y^{-1}$) that is itself a complicated function of atmospheric $O_2$, $CH_4$,
and $CO_2$ (Claire et al., 2006). At each timestep, the distribution of chemical species (e.g., other
than temperature and humidity) in the atmosphere is homogenized (Ridgwell et al., 2007) and $k_{eff}$



is estimated based on the resulting instantaneous mean partial pressures of $O_2$ and $CH_4$ according
to a bivariate fit to a large suite of 1-D atmospheric photochemical models. These photochemical
model results (Claire, *personal communication*) are derived following Claire et al. (2006). Briefly,
values for $k_{eff}$ are computed by a 1-D model of atmospheric photochemistry assuming a range of
fixed surface mixing ratios of $O_2$ and $CH_4$ and a constant atmospheric $CO_2$ of $10^{-2}$ bar. We then fit
a fifth-order polynomial surface to these $k_{eff}$ values as a function of atmospheric $pO_2$ and $pCH_4$
(**Fig. 3**).

Our default parameterization of $O_2$-$O_3$-$CH_4$ chemistry (C06) is fit over a $pO_2$ range of $10^{-14}$ to $10^{-1}$
bar, a $pCH_4$ range of $10^{-6}$ to 2 x $10^{-3}$ bar, and a constant high background $pCO_2$ of $10^{-2}$ bar (Claire
et al., 2006). We thus truncate the atmospheric lifetime of $CH_4$ at a lower bound of 7.6 years in
our default parametrization, and provide an alternative parameterization of photochemical $CH_4$
destruction at roughly modern $pO_2$ and $pCO_2$ (SS03) derived from the results of Schmidt and
Shindell (2003) for use in more geologically recent, high-$O_2$ atmospheres (Reinhard et al., 2017)
(**Fig. 4a**). Although this parameterized photochemistry scheme should represent an improvement
in accuracy relative to that implemented in Olson et al. (2016) (see Daines and Lenton, 2016), it
is important to point out that a range of factors that might be expected to impact the photochemical
destruction rates of $CH_4$ in the atmosphere, including atmospheric $pCO_2$, the atmospheric profile
of $H_2O$, and spectral energy distribution (SED), have not yet been rigorously assessed. Ongoing
model developments in ATCHEM are aimed at implementing a more flexible and inclusive
photochemical parameterization that will allow for robust use across a wider range of atmospheric
compositions and photochemical environments.

As a basic test of our photochemical parameterization, we impose a terrestrial (wetland) flux of
$CH_4$ to the atmosphere (balanced by stoichiometric consumption of $CO_2$ and release of $O_2$), and
allow the oceanic and atmospheric $CH_4$ cycle to spin up for 20 kyr. We then compare steady-state
atmospheric $pCH_4$ as a function of terrestrial $CH_4$ flux to estimates for the last glacial,
preindustrial, and modern periods. Our default parameterization is relatively simple and spans a
very wide range in atmospheric $O_2$ and $CH_4$ inventories. Nevertheless, both the default scheme
and the alternative parameterization for recent geologic history (and analogous planetary
environments) with high-$pO_2$/low-$pCO_2$ atmospheres accurately reproduce atmospheric $pCH_4$



values given estimated glacial, preindustrial, and modern terrestrial $CH_4$ fluxes (**Fig. 4C**), and both
display the predicted saturation of $CH_4$ sinks at elevated atmospheric $CH_4$ observed in more
complex photochemical models. We note, however, the alternative parameterization tends to yield
slightly higher atmospheric $pCH_4$ at surface fluxes greater than ~50 Tmol $y^{-1}$ (**Fig. 4C**). (In the
remainder of the manuscript, we employ the default (C06) parameterization for atmospheric $O_2$-
$O_3$-$CH_4$ chemistry and do not discuss the simple high-$pO_2$/low-$pCO_2$ alternative further.)

### 5. Example applications of the new capabilities in the cGENIE model

#### 5.1. High-$pO_2$ ('modern') steady state

We explore a roughly modern steady state with appropriate continental geography and simulated
overturning circulation (as in Cao et al., 2009) and initialize the atmosphere with $pO_2$, $pCO_2$, and
$pCH_4$ of [v/v] 20.95%, 278 ppm, and 700 ppb, respectively, and with globally uniform oceanic
concentrations of $SO_4^{2-}$ (28 mmol $kg^{-1}$) and $CH_4$ (1 nmol $kg^{-1}$). We fix globally averaged solar
insolation at the modern value (1368 W $m^{-2}$) with seasonally variable forcing as a function of
latitude, and set radiative forcing for $CO_2$ and $CH_4$ equivalent to preindustrial values in order to
isolate the effects of biogeochemistry on steady state tracer distributions. The model is then spun
up for 20 kyr with atmospheric $pO_2$ and $pCO_2$ (and $\delta^{13}C$ of atmospheric $CO_2$) restored to
preindustrial values at every timestep, and with an imposed wetland flux of $CH_4$ to the atmosphere
of 20 Tmol $yr^{-1}$ that has a $\delta^{13}C$ value of -60‰. Atmospheric $pCH_4$ and all oceanic tracers are
allowed to evolve freely.

Surface, benthic, and ocean interior distributions of dissolved oxygen ($O_2$), sulfate ($SO_4^{2-}$), and
methane ($CH_4$) are shown in **Fig. 5** for our roughly modern simulation. Dissolved $O_2$ ([$O_2$])
approaches air saturation throughout the surface ocean, with a distribution that is largely uniform
zonally and with concentrations that increase with latitude as a result of increased solubility at
lower temperature near the poles (**Fig. 5a**). Benthic [$O_2$] shows patterns similar to those expected
for the modern Earth, with relatively high values in the well-ventilated deep North Atlantic, low
values in the deep North Pacific and Indian oceans, and a gradient between roughly air saturation
near regions of deep convection in the high-latitude Atlantic and much lower values in the tropical
and northern Pacific (**Fig. 5d**). Distributions of [$O_2$] in the ocean interior are similar to those of the
modern Earth (**Fig. 5g**) with oxygen minimum zones (OMZs) at intermediate depths underlying





highly productive surface waters, particularly in association with coastal upwelling at low
latitudes.

Concentrations of dissolved $SO_4^{2-}$ ($[SO_4^{2-}]$) are largely invariant throughout the ocean, consistent
with its expected conservative behavior in the modern ocean as one of the most abundant negative
ions in seawater (**Fig. 5**). Slightly higher concentrations in both surface and benthic fields are seen
in association with outflow from the Mediterranean, and are driven by evaporative concentration
(**Fig. 5b**). Benthic $[SO_4^{2-}]$ distributions show some similarity to those of $[O_2]$ (**Fig. 5e**), though
again the differences are very small relative to the overall prescribed initial tracer inventory of 28
mmol kg$^{-1}$ and disappear almost entirely when salinity-normalized (not shown). In the ocean
interior, $[SO_4^{2-}]$ is largely spatially invariant with a value of approximately 28 mmol kg$^{-1}$ (**Fig. 5h**).

Dissolved $CH_4$ concentrations ($[CH_4]$) in the surface and shallow subsurface ocean are much more
variable, but typically on the order of ~1-2 nmol kg$^{-1}$ with slightly elevated concentrations just
below the surface, both of which are consistent with observations from the modern ocean
(Reeburgh, 2007;Scranton and Brewer, 1978). The benthic $[CH_4]$ distribution shows locally
elevated values up to ~300-400 nmol kg$^{-1}$ in shallow regions of the tropical and northern Pacific
and the Indian oceans (**Fig. 5f**), which is also broadly consistent with observations from shallow
marine environments with active benthic $CH_4$ cycling (Jayakumar et al., 2001). Within the ocean
interior, dissolved $CH_4$ can accumulate in the water column in excess of ~100 nmol kg$^{-1}$ in
association with relatively low-$O_2$ conditions at intermediate depths, with zonally averaged values
as high as ~70 nmol kg$^{-1}$ but more typically in the range of ~20-40 nmol kg$^{-1}$ (**Fig. 5i**). These
concentrations are comparable to those observed locally in low-$O_2$ regions of the modern ocean
(Sansone et al., 2001;Chronopoulou et al., 2017;Thamdrup et al., 2019).

**Figure 6** shows the major metabolic fluxes within the ocean's microbial $CH_4$ cycle for our
'modern' configuration. Methanogenesis is focused in regions characterized by relatively low $[O_2]$
and is particularly vigorou s in the Eastern Tropical Pacific, the North Pacific, and the Indian
Ocean (**Fig. 6a**). The highest zonally averaged rates of methanogenesis are observed in northern
tropical and subtropical latitudes, and are focused at a depth of ~1 km (**Fig. 6d**). Rates of microbial
$CH_4$ consumption are generally spatially coupled to rates of methanogenesis, both in a column-



integrated sense (**Fig. 6b, c**) and in the zonal average (**Fig. 6e, f**). This is particularly true for AOM,
rates of which are highest within the core of elevated methanogenesis rates observed in the
northern subtropics. Zonally averaged AOM rates of ~10-15 nmol kg$^{-1}$ d$^{-1}$ compare well with field
measurements of AOM within oceanic OMZs (Thamdrup et al., 2019). In general, the bulk of CH$_4$
produced via microbial methanogenesis is consumed via AOM, either near the seafloor or within
the ocean interior.

**5.2. Low-$p$O$_2$ ('ancient') steady state**
Next, we explore a low-$p$O$_2$ steady state, similar to the Proterozoic Earth (Reinhard et al., 2017)
but played out in a modern continental configuration and overturning circulation, by initializing
the atmosphere with $p$O$_2$, $p$CO$_2$, and $p$CH$_4$ of [v/v] 2.1 x 10$^{-4}$ atm (equivalent to a value 10$^{-3}$ times
the present atmospheric level, PAL), 278 ppm, and 500 ppm, respectively, and globally uniform
oceanic concentrations of SO$_4^{2-}$ (280 μmol kg$^{-1}$) and CH$_4$ (50 μmol kg$^{-1}$). We again fix globally
averaged solar insolation at the modern value (1368 W m$^{-2}$) with seasonally variable forcing as a
function of latitude, and set radiative forcing for CO$_2$ and CH$_4$ equivalent to the modern
preindustrial state in order to isolate the effects of biogeochemistry on steady state tracer
distributions. The model is then spun up for 20 kyr with atmospheric $p$O$_2$ and $p$CO$_2$ (and δ$^{13}$C of
atmospheric CO$_2$) restored to the initial values specified above at every timestep, with an imposed
'geologic' flux of CH$_4$ to the atmosphere of 3 Tmol yr$^{-1}$ at a δ$^{13}$C value of -60‰. Atmospheric
$p$CH$_4$ and all oceanic tracers are allowed to evolve freely.

Surface, benthic, and ocean interior distributions of [O$_2$], [SO$_4^{2-}$], and [CH$_4$], are shown in **Fig. 7**
for our low-$p$O$_2$ simulation. Dissolved O$_2$ concentrations are now extremely heterogeneous
throughout the surface ocean, ranging over an order of magnitude from less than 1 μmol kg$^{-1}$ to
over 10 μmol kg$^{-1}$, with concentrations that are regionally well in excess of air saturation at the
prescribed $p$O$_2$ of 2.1 x 10$^{-4}$ atm (**Fig. 7a**). Previous studies have shown that these features are not
unexpected at very low atmospheric $p$O$_2$ (Olson et al., 2013; Reinhard et al., 2016). We note,
however, that the distribution and maximum [O$_2$] in our low-$p$O$_2$ simulation are both somewhat
different from those presented in Olson et al. (2013) and Reinhard et al. (2016). We attribute this
primarily to the different parameterizations of primary production in the surface ocean. In the
biogeochemical configuration of cGENIE we adopt here, we allow rates of photosynthesis to scale





more directly with available $PO_4^{3-}$ than is the case in these previous studies (Eq. 3), which allows
for higher rates of oxygen production in regions of deep mixing and relatively intense organic
matter recycling below the photic zone (**Fig. 7a**). In any case, as in previous examinations of
surface $[O_2]$ dynamics at low atmospheric $pO_2$ (Olson et al., 2013;Reinhard et al., 2016), our
regional $[O_2]$ patterns still generally track the localized balance between photosynthetic $O_2$ release
and consumption through respiration and reaction with inorganic reductants, rather than
temperature-dependent solubility patterns (**Fig. 5a**). Within the ocean interior, $O_2$ is consumed
within the upper few hundred meters and is completely absent in benthic settings (**Fig. 7d, g**).

In our low-$pO_2$ simulations we initialize the ocean with a globally uniform $[SO_4^{2-}]$ of 280 μmol
kg$^{-1}$, under the premise that marine $SO_4^{2-}$ inventory should scale positively with atmospheric $pO_2$.
With this much lower initial $SO_4^{2-}$ inventory (i.e., $10^2$ times less than the modern ocean), steady
state $[SO_4^{2-}]$ distributions are significantly more heterogeneous than in the modern, high-$pO_2$ case
(**Fig. 7**). Ocean $[SO_4^{2-}]$ is approximately homogeneous spatially in surface waters, even with a
significantly reduced seawater inventory (**Fig. 7a**), but is strongly variable within the ocean interior
(**Fig. 7e, h**). Indeed, in our low-$pO_2$ simulations $SO_4^{2-}$ serves as the principal oxidant for organic
matter remineralization in the ocean interior, with the result that its distribution effectively mirrors
that of $[O_2]$ in the modern case in both spatial texture and overall magnitude (compare **Fig. 7e, h**
with **Fig. 5d, g**). Dissolved $SO_4^{2-}$ in this simulation never drops to zero, a consequence of our initial
280 μmol kg$^{-1}$ concentration of $SO_4^{2-}$ representing the oxidative potential of 560 μmol kg$^{-1}$ of $O_2$,
some 3 times higher than the mean $[O_2]$ value in the modern ocean interior (~170 μmol kg$^{-1}$).

Dissolved $CH_4$ concentrations in the surface and shallow subsurface ocean are variable but much
higher than in our modern simulations, typically on the order of ~1-2 μmol kg$^{-1}$ (**Fig. 7c**). The
benthic $[CH_4]$ distribution shows concentrations up to ~8 μmol kg$^{-1}$, with concentrations in excess
of 1 μmol kg$^{-1}$ pervasively distributed across the seafloor. In general, the benthic $[CH_4]$ distribution
inversely mirrors that of $[SO_4^{2-}]$ (**Fig. 7f**), which results from the fact that in the low-$pO_2$ case
$SO_4^{2-}$ again serves as the principal oxidant of methane. Concentrations of $CH_4$ in the ocean interior
can approach ~10 μmol kg$^{-1}$, but in the zonal average are typically less than 5 μmol kg$^{-1}$ (**Fig. 7i**).
Overall, the oceanic $CH_4$ inventory increases dramatically in the low-$pO_2$ case relative to the
modern simulation, from ~4.5 Tmol $CH_4$ to ~1900 Tmol $CH_4$.






**Figure 8** shows the major metabolic fluxes within the ocean's microbial $CH_4$ cycle for our
'ancient' configuration. Column-integrated rates of microbial methanogenesis are greater than in
the high-$pO_2$ case by up to a factor of ~$10^2$ (**Fig. 8a**), with methanogenesis also showing a much
broader areal distribution. Within the ocean interior, rates of methanogenesis are most elevated in
the upper ~1 km (**Fig. 8d**) as a consequence of elevated rates of organic carbon remineralization
combined with a virtual absence of dissolved $O_2$ beneath the upper ~200 m. Rates of aerobic
methanotrophy, which is effectively absent in the ocean interior (**Fig. 8e**), are elevated relative to
those observed the high-$pO_2$ simulation by less than an order of magnitude and are concentrated
in the tropical surface ocean near the equatorial divergence (**Fig. 8b**). In contrast, AOM is strongly
coupled spatially to microbial methanogenesis, with rates that are often well over ~$10^2$ times higher
than those observed in the high-$pO_2$ case (**Fig. 8c, f**). Once again, AOM dominates the
consumption of $CH_4$ produced in the ocean interior and acts as an extremely effective throttle on
$CH_4$ fluxes to the atmosphere. Despite a significant increase in overall oceanic $CH_4$ burden relative
to our high-$pO_2$ simulation (see above and **Fig. 7i**), atmospheric $pCH_4$ increases only modestly
from ~0.8 ppm to 6 ppm [v/v], equivalent to an additional radiative forcing of only ~2 W m$^{-2}$, due
to efficient microbial consumption in the upper ocean.

**5.3. Atmospheric carbon injection**
To illustrate the capabilities of the model in exploring the time-dependent (perturbation) behavior
of the $CH_4$ cycle, we perform a simple carbon injection experiment in which 3,000 PgC are injected
directly into the atmosphere either as $CH_4$ or as $CO_2$, starting from our modern steady state. The
injection is spread over 1,000 years, with an instantaneous initiation and termination of carbon
input to the atmosphere. The magnitude and duration of this carbon injection, corresponding to 3
PgC y$^{-1}$, is meant to roughly mimic the upper end of estimates for the Paleocene-Eocene Thermal
Maximum, a transient global warming event at ~56 Ma hypothesized to have been driven by
emissions of $CO_2$ and/or $CH_4$ (Kirtland Turner, 2018). This flux is much lower than the current
anthropogenic carbon input of ~10 PgC y$^{-1}$ (Ciais et al., 2013). For simplicity, and because we
focus on only the first 3,000 years following carbon injection, we treat the ocean-atmosphere
system as closed, with the result that all injected carbon ultimately accumulates within the ocean



and atmosphere rather than being removed through carbonate compensation and silicate
weathering.

Following a carbon release to the atmosphere in the form of $CH_4$, there is an immediate and
significant increase in atmospheric $pCH_4$ to values greater than 10 ppmv, followed by a gradual
increase to a maximum of ~12 ppmv throughout the duration of the $CH_4$ input (**Fig. 9a**). Much of
this methane is exchanged with the surface ocean and consumed by aerobic methanotrophy, while
some is photochemically oxidized directly in the atmosphere, both of which lead to a significant
but delayed increase in atmospheric $pCO_2$ (**Fig. 9b**). This increase in atmospheric $pCH_4$ and $pCO_2$
leads to an increase in global average surface air temperature (SAT) of ~7ºC (**Fig. 9d**), an increase
in mean ocean temperature (MOT) of ~2ºC (**Fig. 9e**), along with significant acidification of the
surface ocean (**Fig. 9c**).

The increase in atmospheric $pCO_2$ and drop in ocean pH are nearly identical if we instead inject
the carbon as $CO_2$ rather than $CH_4$. (**Fig. 9b, c**). However, when carbon is injected as $CH_4$, there
is an additional transient increase in global surface air temperature of ~2ºC and roughly 0.5ºC of
additional whole ocean warming for the same carbon input and duration (**Fig. 9f**). This results
from the fact that mole-for-mole, $CH_4$ is a much more powerful greenhouse gas than is $CO_2$, and
oxidation of $CH_4$ to $CO_2$ is not instantaneous during the carbon release interval. Combined, these
factors result in a disequilibrium situation in which a proportion of carbon released to the
atmosphere remains in the form of $CH_4$ rather than $CO_2$, providing an enhancement of warming,
especially during the duration of carbon input. This warming enhancement should be considered
in past events during which $CH_4$ release is suspected as a key driver of warming. For instance,
additional warming due to $CH_4$ forcing may help explain the apparent discrepancy between the
amount of warming reconstructed by proxy records and proposed carbon forcing during the PETM
(Zeebe et al., 2009)

**5.4. Atmospheric $pCH_4$ on the early Earth**
Using our low-$pO_2$ steady state as a benchmark case (**Section 5.2**), we briefly explore the
sensitivity of atmospheric $pCH_4$ to a subset of model variables. All model ensembles are initially
configured with globally homogeneous marine $SO_4^{2-}$ and $CH_4$ inventories and a background



geologic $CH_4$ flux of 3 Tmol $y^{-1}$, and are spun up for 20 kyr with a fixed $pO_2$ and $pCO_2$. We report
atmospheric $pCH_4$ from the final model year. Our purpose here is not to be exhaustive or to
elucidate any particular period of Earth history, but to demonstrate some of the major factors
controlling the atmospheric abundance of $CH_4$ on a low-oxygen Earth-like planet. We present
results from individual sensitivity ensembles from our benchmark low-$pO_2$ case over the following
parameter ranges: (1) atmospheric $pO_2$ between $10^{-4}$ to $10^{-1}$ times the present atmospheric level
(PAL), equivalent to roughly $2 \times 10^{-5}$ and $2 \times 10^{-2}$ atm, respectively; (2) initial marine $SO_4^{2-}$
inventories corresponding to globally uniform seawater concentrations between 0 and 1,000 µmol
$kg^{-1}$; and (3) biological energy quanta (BEQ) for anaerobic methane oxidation between 5 and 30
kJ $mol^{-1}$.

Results for our low-$pO_2$ sensitivity ensembles are shown in **Figure 10**. We find a similar sensitivity
of atmospheric $pCH_4$ to atmospheric $pO_2$ to that observed by (Olson et al., 2016). In particular,
atmospheric $CH_4$ abundance initially increases as atmospheric $pO_2$ drops below modern values to
roughly 2-3% PAL, after which decreasing $pO_2$ causes $pCH_4$ to drop. This behavior is well-known
from previous 1-D photochemical model analysis, and arises principally from increasing
production of OH via water vapor photolysis as shielding of $H_2O$ by ozone ($O_3$) decreases at low
atmospheric $pO_2$ (Pavlov et al., 2003;Claire et al., 2006;Goldblatt et al., 2006). However, peak
atmospheric $pCH_4$ is significantly reduced in our models relative to those of Olson et al. (2016).
For example, at an 'optimal' atmospheric $pO_2$ of ~2.5% PAL Olson et al. (2016) predict a steady
state atmospheric $pCH_4$ of ~35 ppmv, while we predict a value of ~10 ppmv (**Fig. 10a**). This
difference can be attributed to our updated $O_2$-$O_3$-$CH_4$ photochemistry parameterization together
with a significant upward revision in the rate constant for aerobic methanotrophy. Nevertheless,
our results strongly reinforce the arguments presented in Olson et al. (2016), and taken at face
value further marginalize the role of $CH_4$ as a significant climate regulator at steady state during
most of the Proterozoic Eon (between ~2.5 and 0.5 Ga).

Atmospheric $CH_4$ abundance is also strongly sensitive to the marine $SO_4^{2-}$ inventory (**Fig. 10b**).
The scaling we observe between initial $SO_4^{2-}$ inventory and steady state atmospheric $pCH_4$ is very
similar to that reported by Olson et al. (2016), with a sharp drop in the marine $CH_4$ inventory and
atmospheric $CH_4$ abundance as marine $SO_4^{2-}$ drops below ~100 µmol $kg^{-1}$ (**Fig. 10b**). The





implication is that for most of Earth history anaerobic oxidation of $CH_4$ in the ocean interior has
served as an important inhibitor of $CH_4$ fluxes from the ocean biosphere. However, during much
of the Archean Eon (between 4.0 and 2.5 Ga), sulfur isotope analysis indicates that marine $SO_4^{2-}$
concentrations may instead have been on the order of ~1-10 μmol kg$^{-1}$ (Crowe et al., 2014), while
atmospheric $pO_2$ would also have been much lower than the values examined here (Pavlov &
Kasting, 2002). The impact of the ocean biosphere and redox chemistry on atmospheric $pCH_4$ and
Earth's climate system may thus have been much more important prior to ~2.5 billion years ago.

Interestingly, atmospheric $CH_4$ is significantly impacted by the value chosen for the biological
energy quantum (BEQ). With all other parameters held constant, we observe an increase in steady
state atmospheric $pCH_4$ from ~7 ppmv to ~25 ppmv when increasing the BEQ value from 20 to
30 kJ mol$^{-1}$ (**Fig. 10c**). This effect is mediated primarily by the importance of anaerobic
methanotrophy when atmospheric $pO_2$ is low and the ocean interior is pervasively reducing. The
standard free energy of AOM is of the same order of magnitude as the BEQ (see above), which
elevates the importance of thermodynamic drive in controlling global rates of AOM. We would
expect this effect to be much less important when aerobic methanotrophy is the predominant $CH_4$
consuming process within the ocean biosphere, as the standard free energy of this metabolism is
over an order of magnitude greater than typical BEQ values for microbial metabolism (e.g.,
Hoehler, 2004). In any case, our results suggest that the role of thermodynamics should be borne
in mind in scenarios for which AOM is an important process in the $CH_4$ cycle and seawater [$SO_4^{2-}$
] is relatively low.

**6. Discussion and Conclusions**
The global biogeochemical cycling of $CH_4$ is central to the climate and redox state of planetary
surface environments, and responds to the internal dynamics of other major biogeochemical cycles
across a very wide range of spatial and temporal scales. There is thus strong impetus for the
ongoing development of a spectrum of models designed to explore planetary $CH_4$ cycling, from
simple box models to more computationally expensive 3-D models with dynamic and interactive
ocean circulation. Our principal goal here is the development of a mechanistically realistic but
simple and flexible representation of $CH_4$ biogeochemical cycling in Earth's ocean-atmosphere
system, with the hope that this can be further developed to explore steady state and time-dependent



changes to global $CH_4$ cycle in Earth's past and future and ultimately to constrain $CH_4$ cycling
dynamics on Earth-like planets beyond our solar system.

To accomplish this, we have refined the organic carbon remineralization scheme in the cGENIE
Earth system model to reflect the impact of anaerobic organic matter recycling in sinking
aggregates within oxygenated waters, and to include the carbon cycling and isotopic effects of
microbial $CH_4$ production. We have also incorporated revised schemes for microbial $CH_4$
consumption that include both kinetic and thermodynamic constraints, and have updated the
parameterized atmospheric $O_2$-$O_3$-$CH_4$ photochemistry to improve accuracy and for use across a
wider range of atmospheric $pO_2$ values than that explored in previous work. Simulations of roughly
modern (high-$O_2$) and Proterozoic (low-$O_2$) Earth system states demonstrate that the model
effectively reproduces the first-order features of the modern ocean-atmosphere $CH_4$ cycle, and can
be effectively implemented across a wide range of atmospheric $O_2$ partial pressures and marine
$SO_4^{2-}$ concentrations. In addition, our results strongly reinforce the conclusions of Olson et al.
(2016) for the Proterozoic Earth system, while going beyond this to posit that the thermodynamics
of anaerobic $CH_4$ consumption may have been important in regulating atmospheric $CH_4$ abundance
during Archean time. Finally, our simulation of PETM-like carbon injection demonstrates the
importance of explicitly considering $CH_4$ radiative forcing during transient warming events in
Earth history.

We suggest that ongoing and future development work should focus on: (1) more rigorous tuning
of organic carbon remineralization and $CH_4$ production/consumption schemes based on data fields
from the modern ocean; (2) development and implementation of a more flexible parameterization
of atmospheric photochemistry that allows the roles of atmospheric temperature structure, water
vapor abundance, and atmospheric $pCO_2$ to be explored; (3) coupling of deep ocean chemistry
with a description of marine methane hydrates and associated sedimentary $CH_4$ cycling; and (4)
developing a representation of the production/consumption of $CH_4$ by terrestrial ecosystems.






**7. Model code availability**

A manual describing code installation, basic model configuration, and an extensive series of tutorials is provided. The Latex source of the manual and PDF file can be obtained by cloning (https://github.com/derpycode/muffindoc). The user manual contains instructions for obtaining, installing, and testing the code, as well as running experiments. The version of the code used in this paper is tagged as release v0.9.10 and has a DOI of 10.5281/zenodo.3620846. Configuration files for the specific experiments presented in the paper can be found in: cgenie.muffin/genie-userconfigs/MS/reinhardetal.GMD.2020. Details of the different experiments, plus the command line needed to run each, are given in README.txt.

**Author contributions:**

CTR, SLO, and AR developed new model code. CTR and CP compiled and analyzed empirical data for rates of methanotrophy. CTR performed all model simulations and data analysis. CTR prepared the manuscript with contributions from all co-authors.

**Acknowledgements:**

CTR acknowledges support from the NASA Astrobiology Institute (NAI), the Alfred P. Sloan Foundation, and the NASA Nexus for Exoplanet System Science (NExSS). SLO acknowledges support from the T.C. Chamberlin Postdoctoral Fellowship in the Department of Geophysical Sciences at the University of Chicago. AR, SKT, and YK were supported in part by an award from the Heising-Simons Foundation. We also thank Mark Claire for providing unpublished photochemical model results.





**TABLES:**

**Table 1.** Default parameters for organic matter production and water column remineralization.

| parameter | description | default value | units | source |
|---|---|---|---|---|
| *uptake/photosynthesis* | | | | |
| $\lambda$ | rate constant for DOM degradation | 0.5 | $y^{-1}$ | 1 |
| $\upsilon$ | fractional partitioning into DOM | 0.66 | — | 1 |
| $\tau_{bio}$ | nutrient uptake timescale | 63 | d | 2 |
| $\kappa_I$ | light limitation term | 20 | W m$^{-2}$ | 3 |
| $\kappa_P$ | half-saturation constant for PO$_4$ uptake | 2.1 x 10$^{-7}$ | mol kg$^{-1}$ | 1 |
| $k_{T0}$ | pre-exponential temperature constant | 0.59 | — | [see text] |
| $k_{eT}$ | exponential temperature constant | 15.8 | — | [see text] |
| *organic remineralization* | | | | |
| $r_1^{POM}$ | partitioning into labile POM fraction | 0.945 | — | 1 |
| $l_1^{POM}$ | *e*-folding depth for labile POM fraction | 589 | m | [see text] |
| $r_2^{POM}$ | partitioning into refractory POM fraction | 0.055 | — | 4 |
| $l_2^{POM}$ | *e*-folding depth for recalcitrant POM fraction | 10$^6$ | m | [see text] |
| $\kappa_{O_2}$ | half-saturation constant for aerobic respiration | 1.0 x 10$^{-7}$ | mol kg$^{-1}$ | [see text] |
| $\kappa_{O_2}^i$ | inhibition constant for aerobic respiration | 1.0 x 10$^{-6}$ | mol kg$^{-1}$ | [see text] |
| $\kappa_{SO_4}$ | half-saturation constant for sulfate reduction | 5.0 x 10$^{-4}$ | mol kg$^{-1}$ | 4 |
| $\kappa_{SO_4}^i$ | Inhibition constant for sulfate reduction | 5.0 x 10$^{-4}$ | mol kg$^{-1}$ | 4 |

[1]Ridgwell et al. (2007); [2]Meyer et al. (2016); [3]Doney et al. (2006); [4]Olson et al. (2016)








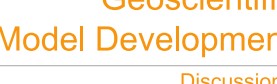 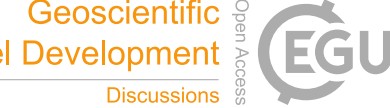

**Table 2.** Default kinetic and thermodynamic parameters for oceanic methane cycling. Activity coefficients
are estimated for $T = 25^\circ C$ and $S = 35‰$.

| parameter | description | default value | units | source |
|---|---|---|---|---|
| *kinetic parameters* | | | | |
| $k_{AER}$ | rate constant for aerobic methanotrophy | 0.10 | $y^{-1}$ | [see text] |
| $\kappa_O^{AER}$ | half-saturation constant for $O_2$ | $2.0 \times 10^{-5}$ | $mol\ kg^{-1}$ | [see text] |
| $k_{AOM}$ | rate constant for AOM | 0.01 | $y^{-1}$ | [see text] |
| $\kappa_S^{AOM}$ | AOM half-saturation constant for $SO_4^{2-}$ | $5.0 \times 10^{-4}$ | $mol\ kg^{-1}$ | 1 |
| *thermodynamic parameters* | | | | |
| $\Delta G_{r,AER}^0$ | standard free energy yield of aerobic methanotrophy | -858.967 | $kJ\ mol^{-1}$ | 2 |
| $\Delta G_{r,AOM}^0$ | standard free energy yield of AOM | -33.242 | $kJ\ mol^{-1}$ | 2 |
| $\Delta G_{BQ,AER}$ | minimum free energy for aerobic methanotrophy | -15.0 | $kJ\ mol^{-1}$ | [see text] |
| $\Delta G_{BQ,AOM}$ | minimum free energy for AOM | -15.0 | $kJ\ mol^{-1}$ | 2-5 |
| $\gamma_{CH_4}$ | activity coefficient for dissolved $CH_4$ | 1.20 | — | 6-8 |
| $\gamma_{CO_2}$ | activity coefficient for aqueous $CO_2$ | 1.17 | — | 9 |
| $\gamma_{O_2}$ | activity coefficient for dissolved $O_2$ | 1.14 | — | 10 |
| $\gamma_{HCO_3^-}$ | activity coefficient for dissolved $HCO_3^-$ | 0.58 | — | 11, 12 |
| $\gamma_{HS^-}$ | activity coefficient for dissolved $HS^-$ | 0.75 | — | 13 |
| $\gamma_{SO_4^{2-}}$ | activity coefficient for dissolved $SO_4^{2-}$ | 0.10 | — | 11 |
| $R$ | gas constant | $8.2144 \times 10^{-3}$ | $kJ\ K^{-1}\ mol^{-1}$ | |
| $\chi$ | stoichiometric number | 1.0 | — | 14 |
| *isotopic parameters* | | | | |
| $\varepsilon_{CH_4}$ | methanogenesis isotope effect | -35.0 | ‰ | [see text] |

[1]Olson et al. (2016); [2]Regnier et al. (2011); [3]Schink (1997); [4]Hoehler et al. (2001); [5]Hoehler (2004); [6]Stoessell and
Byrne (1982); [7]Cramer (1984); [8]Duan et al. (1992); [9]Johnson (1982); [10]Clegg and Brimblecombe (1990); [11]Ulfsbo et
al. (2015); [12]Berner (1965); [13]Helz et al. (2011); [14]Dale et al. (2008)



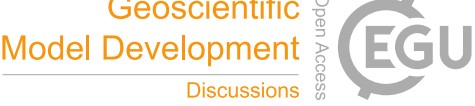

**Table 3.** Default constants and coefficients for CH$_4$ gas exchange. All default parameter values derived
from Wanninkhof (1992). Schmidt number coefficients are for $S = 35‰$.

| parameter | description | default value |
|---|---|---|
| $a_1$ | Bunsen temperature coefficient 1 | -68.8862 |
| $a_2$ | Bunsen temperature coefficient 2 | 101.4956 |
| $a_3$ | Bunsen temperature coefficient 3 | 28.7314 |
| $b_1$ | Bunsen salinity coefficient 1 | -0.076146 |
| $b_2$ | Bunsen salinity coefficient 2 | 0.043970 |
| $b_3$ | Bunsen salinity coefficient 3 | -0.0068672 |
| $c_1$ | Schmidt temperature coefficient 1 | 2039.2 |
| $c_2$ | Schmidt temperature coefficient 2 | 120.31 |
| $c_3$ | Schmidt temperature coefficient 3 | 3.4209 |
| $c_4$ | Schmidt temperature coefficient 4 | 0.040437 |
| $k$ | Gas exchange constant | 0.31 |





**FIGURES:**

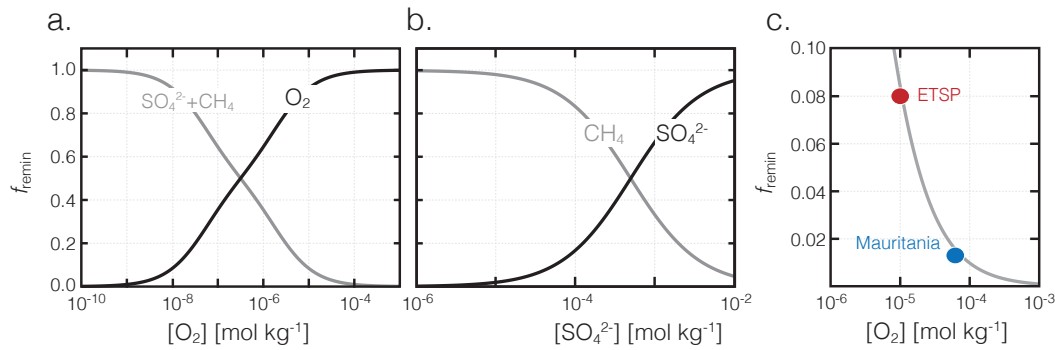

**Figure 1.** Fractional organic carbon remineralization by aerobic respiration, sulfate reduction, and methanogenesis in
our modified organic matter remineralization scheme. In (a), relative rates of aerobic ($O_2$) and anaerobic ($SO_4^{2-} + CH_4$)
remineralization are plotted as a function of dissolved [$O_2$]. In (b), relative anaerobic remineralization rates are
partitioned between sulfate reduction and methanogenesis as a function of dissolved [$SO_4^{2-}$] (dissolved [$O_2$] is held
constant at $10^{-10}$ mol kg$^{-1}$). Shown in (c) are our estimated anaerobic remineralization fractions (grey curve) compared
to estimates from a particle biogeochemical model applied to oxygen minimum zones (OMZs) in the Eastern Tropical
South Pacific (ETSP) and Mauritanian upwelling (Bianchi et al., 2018).


















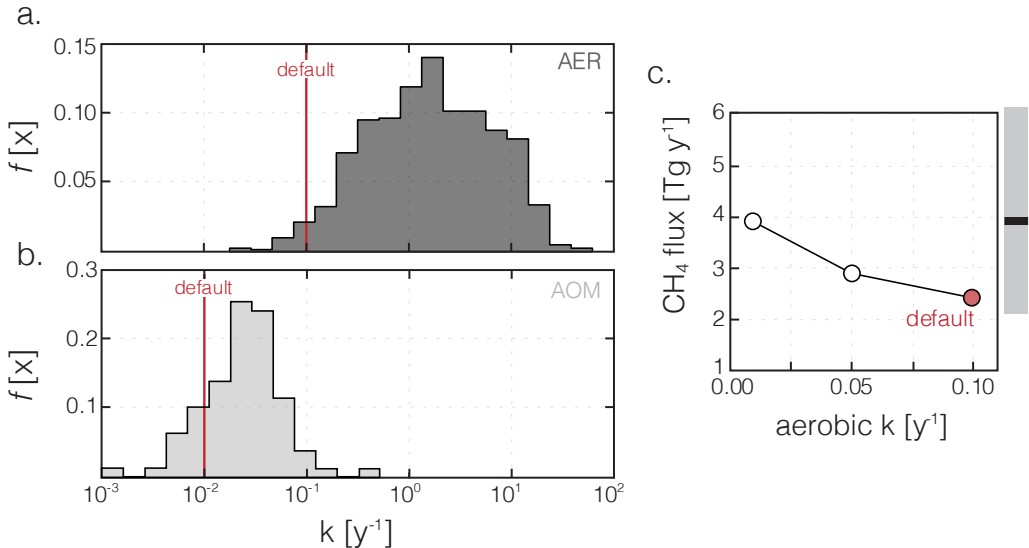


**Figure 2.** Compilation of rate constants for aerobic (AER; a) and anaerobic (AOM; b) methane oxidation. Rate
constants are corrected for *in situ* temperature using a $Q_{10}$ of 2 (see Supplementary Materials). Vertical red lines show
our default values as reported in **Table 2**. Shown in (c) are globally integrated diffusive fluxes of $CH_4$ from the ocean
for a range of rate constants for aerobic methanotrophy, including our default simulation. The bar to the right of (c)
shows the median (black bar) and 90% credible interval (grey shading) for estimates of the modern oceanic diffusive
flux from (Weber et al., 2019)















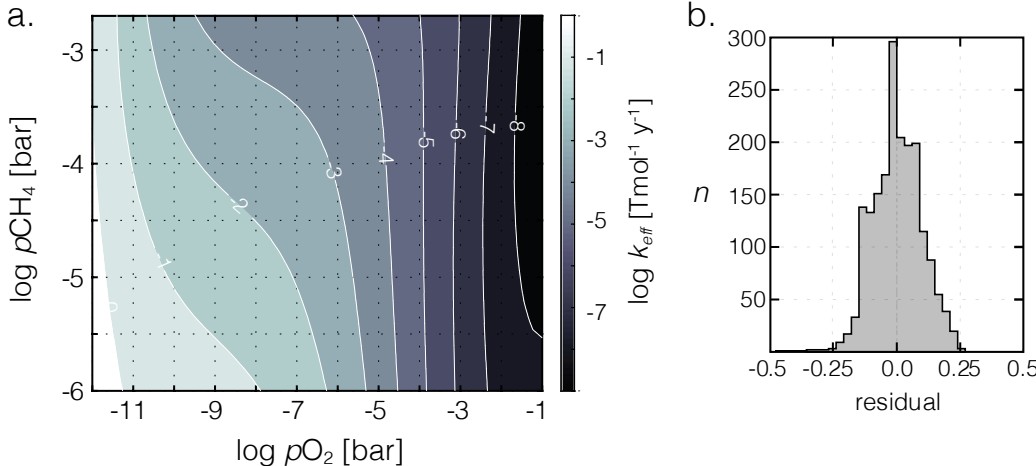

**Figure 3.** Shown in (a) is the bivariate fit to a suite of 1-D atmospheric photochemical runs for the effective rate constant ($k_{eff}$) parameterizing $O_2$-$O_3$-$CH_4$ photochemistry in ATCHEM. Shown in (b) is a frequency distribution of the residuals on $k_{eff}$ from the underlying photochemical model output.

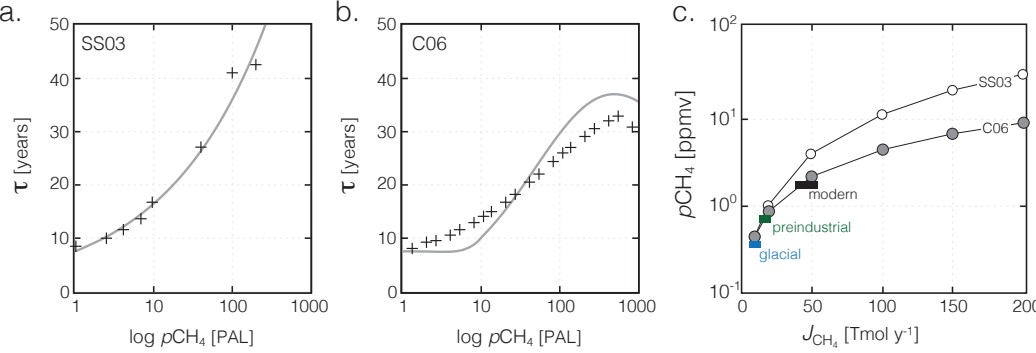

830

**Figure 4.** Comparison of steady-state atmospheric $pCH_4$ as a function of terrestrial $CH_4$ flux with modern/recent
estimates. Shown in (a) is an exponential fit to the 2-D photochemistry model of Schmidt and Shindell (2003) (SS03),
with individual model runs shown as black crosses. Shown in (b) is a plane through the bivariate fit shown in Figure
3 (grey curve), compared with the ensemble of 1-D atmospheric photochemical models at $pO_2 = 0.1$ atm (black
crosses; see text). Shown in (c) are steady-state atmospheric $CH_4$ values as a function of imposed terrestrial $CH_4$ flux
in our 'modern' configuration (circles), compared to estimates for the glacial, preindustrial, and modern $CH_4$ cycles
(Kirschke et al., 2013;Bock et al., 2017;Paudel et al., 2016)




















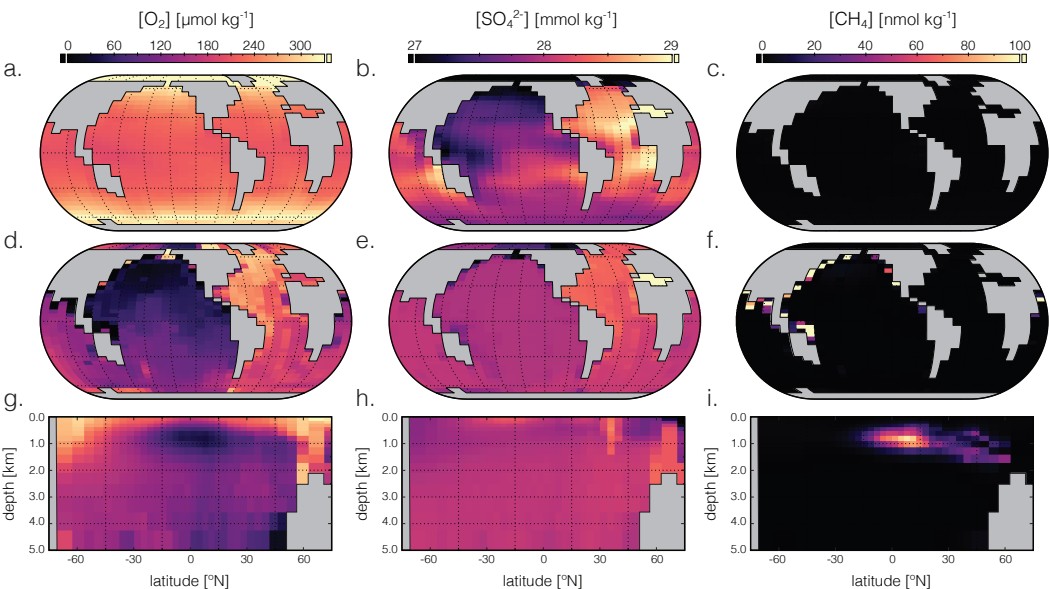

**Figure 5.** Tracer distributions in surface (a-c) and benthic (d-f) grid cells and in the zonally averaged ocean interior (g-i) for $O_2$ (a, d, g), $SO_4^{2-}$ (b, e, h), and $CH_4$ (c, f, i) in our 'modern' configuration. Note different concentration units for each tracer.



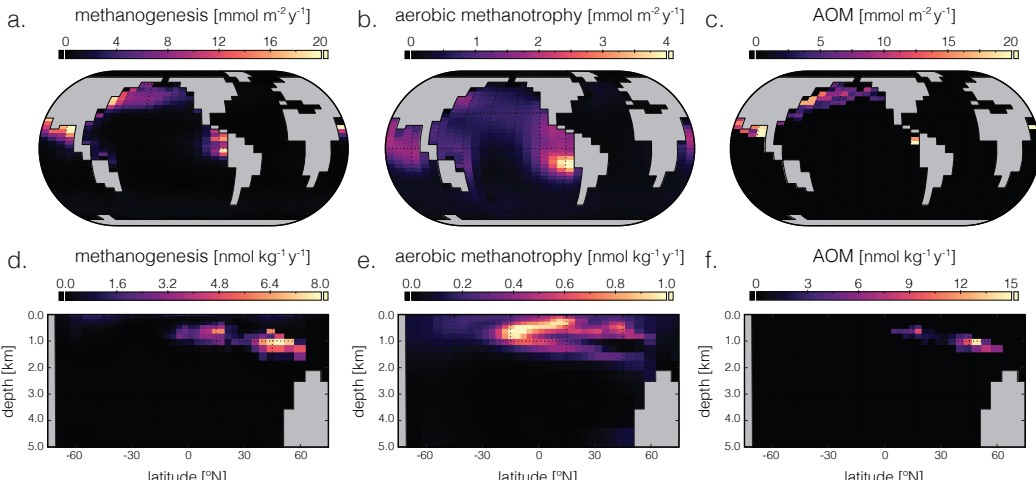


**Figure 6.** Major biological fluxes in the marine methane cycle for our 'modern' configuration. Panels show column
integrated (a-c) and zonally averaged (d-f) rates of methanogenesis, aerobic methanotrophy, and anaerobic methane
oxidation (AOM) in the ocean interior.























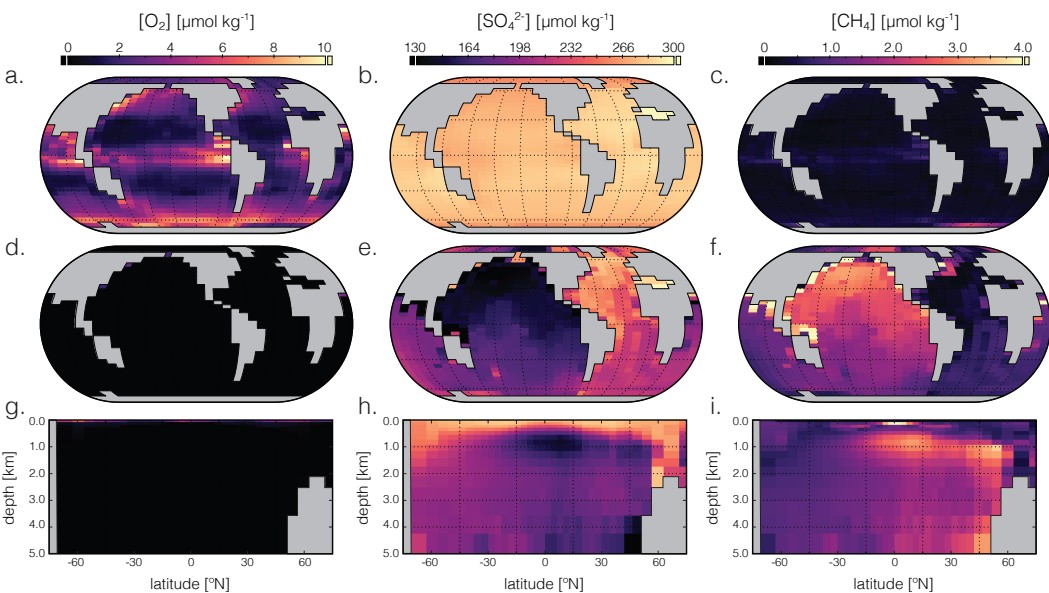

**Figure 7.** Tracer distributions in surface (a-c) and benthic (d-f) grid cells and in the zonally averaged ocean interior (g-i) for $O_2$ (a, d, g), $SO_4^{2-}$ (b, e, h), and $CH_4$ (c, f, i) in our 'ancient' configuration (see text). Note different concentration units for each tracer, and the differing scales relative to Figure 5.



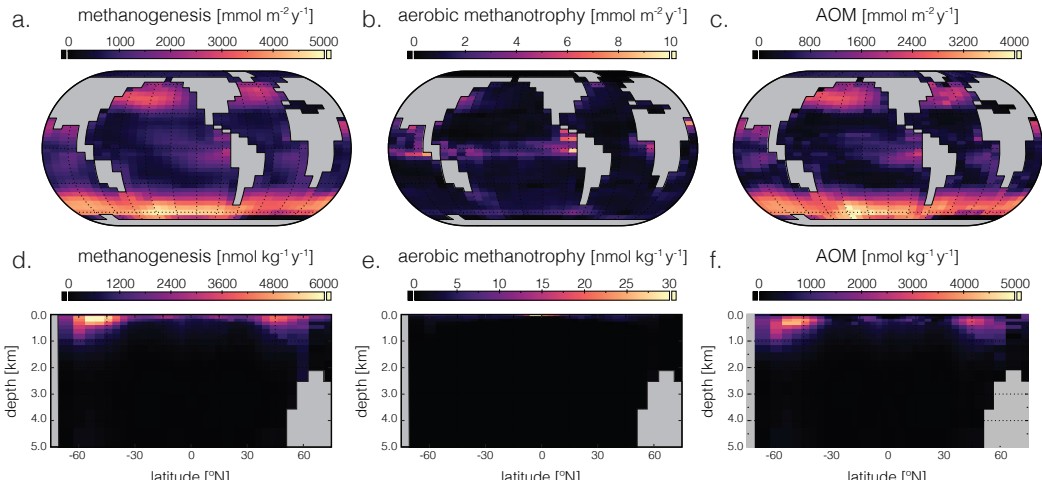

**Figure 8.** Major biological fluxes in the marine methane cycle for our 'ancient' configuration. Panels show column integrated (a-c) and zonally averaged (d-f) rates of methanogenesis, aerobic methanotrophy, and anaerobic methane oxidation (AOM) in the ocean interior.



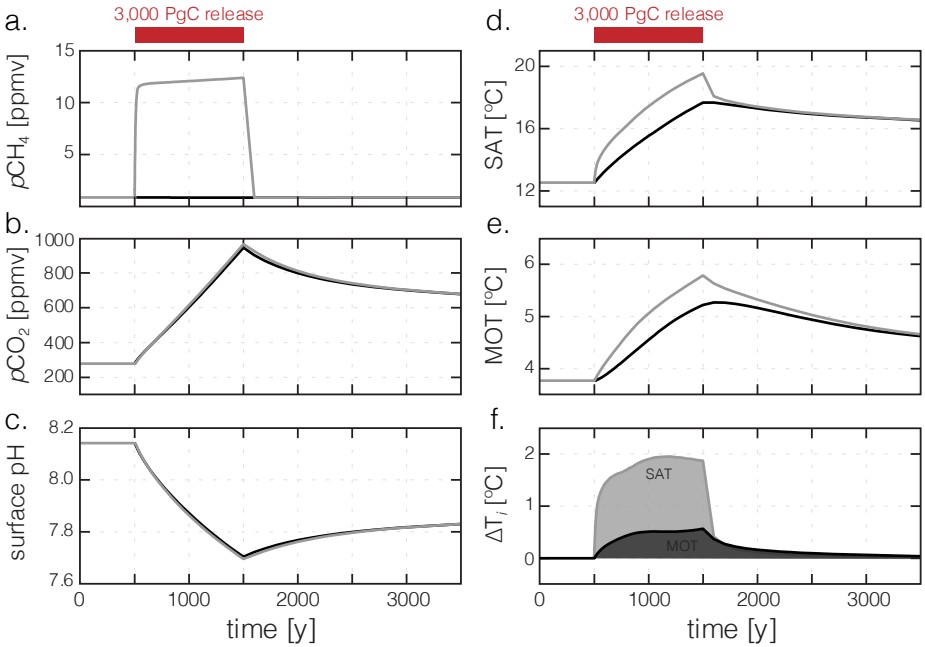

**Figure 9.** Response to a 3,000 PgC release directly to the atmosphere spread over 1,000 years, assuming carbon is
injected as either $CH_4$ or $CO_2$. Atmospheric $pCH_4$ (a), $pCO_2$ (b), mean surface ocean pH (c), mean surface air
temperature (SAT; d), and mean ocean temperature (MOT; e) are shown for a $CH_4$ injection (grey) and a $CO_2$ injection
(black). Panel (f) shows the difference in SAT and MOT between the $CH_4$ and $CO_2$ injection scenarios ($\Delta T_i = T_{CH4,i} -$
$T_{CO2,i}$) through time.



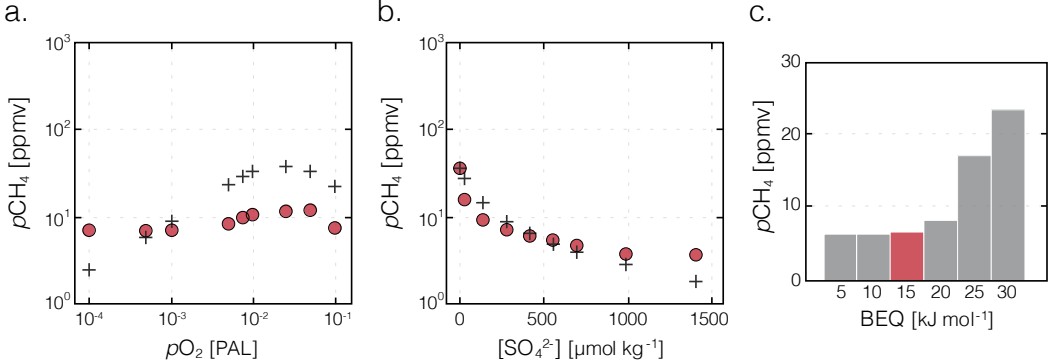

**Figure 10.** Sensitivity ensembles of our 'ancient' configuration compared to the results of Olson et al. (2016). Steady-state atmospheric $pCH_4$ values as a function of assumed atmospheric $pO_2$ (a) and initial marine $SO_4^{2-}$ inventory (b) are shown for our 'ancient' configuration (filled circles; see text) and from Olson et al. (black crosses). Shown below are additional ensembles showing the impact of varying the minimum free energy yield required for microbial methane oxidation (BEQ; c) on atmospheric $pCH_4$. All simulations were spun up from cold for 20 kyr, with the results shown from the last model year.





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
