# Peer review of "Oceanic and atmospheric methane cycling in the cGENIE Earth system model"

_Geoscientific Model Development, 2020_

## Short Comment (SC1) · 31 Mar 2020

Dear authors,

in my role as Executive editor of GMD, I would like to bring to your attention our Editorial version 1.2:

https://www.geosci-model-dev.net/12/2215/2019/

This highlights some requirements of papers published in GMD, which is also available on the GMD website in the 'Manuscript Types' section:

http://www.geoscientific-model-development.net/submission/manuscript_types.html

In particular, please note that for your paper, the following requirement has not been

met in the Discussions paper:

- "The main paper must give the model name and version number (or other unique identifier) in the title."

Please add a version number (the Github TAG number) for cGENIE in the title upon your revised submission to GMD.

Yours,

Astrid Kerkweg

---

## Referee Comment (RC1) · David Archer (Referee) · 6 Apr 2020

This paper describes a model of the global methane cycle, from the sea floor through the ocean to the top of the atmosphere. It is a somewhat new scope for modeling methane than I have seen before (I was not familiar with the cited Olson predecessor model). Papers like this are fun to read because the authors have to figure out what the most important processes are, what to put in and what to leave out, and also what the best tests of the simulation should be. Then, it's interesting to see what the sensitivities of the model are, a chance for a model to teach us something. In this paper, it is interesting to see how the concentrations of O2 and SO4 affect the distribution and cycling of CH4, and interesting that a significant amount of a released CH4 spike should degrade in the oceans, as two examples. This paper is is a useful

contribution which could be published nearly as is.

The paper is very clearly written. I have just a few specific comments and suggestions.

The kinetics of CH4 degradation are described as an O2-O3-CH4 parameterization, but there is no mention of O3 except for that. Does O3 do anything interesting at different O2 concentrations, or during the CH4 spike? If not, it would still be worth a sentence describing what role O3 is playing in the parameterization, just for clarity.

On line 601 it is suggested that CH4 warming might explain the warmth of the PETM. This was what Schmidt and Shindell assumed, but it doesn't work because the warming persisted after the release period was over, meaning that it must have been CO2, not CH4.

---

## Referee Comment (RC2) · Jeff Ridley (Referee) · 6 Jul 2020

This paper describes the methane cycle in a model of intermediate complexity. This simple model seems to be necessary due to the long spin-up times of the reservoirs. The paper is made more awkward than it needs to be by trying to simulate the PET and primitive Earth. This is done with little reference and comparison to even simpler models. This is a well written paper, is suitable for GMD and can be accepted as is with the following minor suggestions:

478: Restructure The subject of this sentence should be 'metabolic fluxes' not 'Figure 6'

547: Again, making the Figure lead the science, rather than vice versa.

558: suggest rephrasing for international audiences 'effective throttle'

646: Avoid telling the reader that they should find something 'interesting'.

---

## Author Comment (AC1) · 27 Aug 2020

Dear Editor and Reviewers,

We thank the reviewers and executive editor for their constructive suggestions and comments on our manuscript. Below is a point-by-point response to all reviewer and editor comments.

Many thanks,

Chris Reinhard (on behalf of all coauthors)

EXECUTIVE EDITOR COMMENTS:

In particular, please note that for your paper, the following requirement has not been

met in the Discussions paper:

"The main paper must give the model name and version number (or other unique identifier) in the title." Please add a version number (the Github TAG number) for cGENIE in the title upon your revised submission to GMD.

—This information has now been added to the title.

REVIEWER #1

The kinetics of CH4 degradation are described as an O2-O3-CH4 parameterization, but there is no mention of O3 except for that. Does O3 do anything interesting at different O2 concentrations, or during the CH4 spike? If not, it would still be worth a sentence describing what role O3 is playing in the parameterization, just for clarity.

—This is a good point. We have added the following text to clarify: "We note that in this parameterization, O3 abundance is not calculated explicitly, but rather the photochemical destruction rate of CH4 in the atmosphere is controlled by the combined atmospheric chemistry implicitly embedded within keff (Claire et al., 2006; Goldblatt et al., 2006)."

On line 601 it is suggested that CH4 warming might explain the warmth of the PETM. This was what Schmidt and Shindell assumed, but it doesn't work because the warming persisted after the release period was over, meaning that it must have been CO2, not CH4.

—We certainly do not mean to suggest that the temperature changes observed during the course of the PETM are entirely attributable to changes in CH4 cycling. The time-dependent analysis is only meant to illustrate the transient behavior of the model during an idealized perturbation, rather than to evaluate any particular scenario for explaining previous climate transients in Earth's history. We have added the following clause to this portion of the text in order to emphasize this (Line 573): "This is meant only to illustrate the time-dependent behavior of the model in the face of an idealized

carbon cycle perturbation, rather than to evaluate any particular scenario for explaining previous climate transients in Earth's history."

REVIEWER #2

478: Restructure The subject of this sentence should be 'metabolic fluxes' not 'Figure 6'

—This has been changed.

547: Again, making the Figure lead the science, rather than vice versa.

—This has been changed.

558: suggest rephrasing for international audiences 'effective throttle'

—This has been changed to: "...AOM dominates the consumption of CH4 produced in the ocean interior and is extremely effective at reducing CH4 fluxes to the atmosphere."

646: Avoid telling the reader that they should find something 'interesting'.

—We have removed "Interestingly, ..." from the beginning of the sentence.

Please also note the supplement to this comment:
https://gmd.copernicus.org/preprints/gmd-2020-32/gmd-2020-32-AC1-supplement.zip
* * *